# Flow Matching for 3D Craniofacial Skeletal Data Generation

**Giacomo Melacini**[1]                                                 GIACOMO.MELACINI2@UNIBO.IT
**Stefano Mazzocchetti**[2]                                    STEFANO.MAZZOCCHETT5@UNIBO.IT
**Giuseppe Lisanti**[1]                                              GIUSEPPE.LISANTI@UNIBO.IT
**Luigi Di Stefano**[1]                                            LUIGI.DISTEFANO@UNIBO.IT
**Samuele Salti**[1]                                                  SAMUELE.SALTI@UNIBO.IT
[1] *CVLAB, University of Bologna*
[2] *DIMESLab, University of Bologna*

**Editors:** Accepted for publication at MIDL 2026

## Abstract

In the medical domain, the use of Machine Learning (ML) techniques for diagnosis, treatment planning, and medical imaging interpretation is becoming increasingly important. However, these approaches require a large amount of data, which is challenging to access due to its sensitive nature and related privacy concerns. Synthetic data generation, enabled by advances in generative techniques, provides a solution to create large anonymized datasets for training models without compromising patient privacy. Recently, Flow Matching with Optimal Transport (OTFM) has proven to be an effective technique for generating realistic 2D natural images, surpassing existing methods, but its usage for 3D medical data generation is limited. In this work we generate craniofacial skeletal data using OTFM and test the validity of the results in two clinical downstream tasks: skull alignment and shape completion. Moreover, we compare the quality of synthetic data generated with OTFM with the ones generated using Denoising Diffusion Probabilistic Models (DDPMs). We show that Flow Matching with Optimal Transport is an effective technique for generating synthetic data and that, in this context, it outperforms DDPMs both in quality and robustness.

**Keywords:** Generative modelling, 3D, Flow Matching, Diffusion Models, Head CT.

## 1. Introduction

In the medical domain, data plays a crucial role in diagnosis, treatment planning, and research. In recent years, Machine Learning (ML) models trained on such data have demonstrated strong potential in supporting clinical decisions, predicting disease outcomes, personalizing treatment plans or assisting in medical imaging interpretation (Javaid et al., 2022). However, these models often require large volumes of data, which can be difficult to obtain in practice. Medical data sharing is constrained by strict regulations, which vary across different jurisdictions. Notable examples include the EU General Data Protection Regulation (GDPR), the US Health Insurance Portability and Accountability Act (HIPAA) and China's Personal Information Protection Law (PIPL). Moreover, the creation of large medical datasets is also inhibited by the high cost of annotations, which require expert clinicians and significant time investment. To address these challenges, deep generative models can be used to generate synthetic medical data (Han et al., 2020; Kazerouni et al., 2023). In addition to data scarcity, another issue with using ML models to address clinical tasks lies in

the unbalanced nature of medical datasets. The presence of unbalanced data distributions can introduce bias and limit the generalizability of the trained models. However, balanced data are often not available because the number of data samples for a specific event, such as a disease or a particular medical examination, depends on how frequently that event occurs, which can vary significantly. Similarly, medical datasets rarely cover uniformly relevant clinical features such as age, sex, ethnicity and disease stage. Generative models, when trained with conditioning, could be used to reduce this problem by synthesizing additional data for under-represented classes (Li et al., 2024).

Different generative techniques have been developed to advance image generation, including Variational Autoencoders (VAE) (Kingma and Welling, 2014), Generative Adversarial Networks (GANs) (Goodfellow et al., 2020), and Denoising Diffusion Probabilistic Models (DDPMs) (Ho et al., 2020). VAEs offer an explicit probabilistic formulation but are known to produce blurry images; GANs rely on adversarial training to produce high quality samples, but suffer from mode collapse (Che et al., 2017); DDPMs learn to reverse a diffusion process to generate images and, when trained in the latent space of an Autoencoder (Rombach et al., 2022), enable high-resolution image synthesis. Recently, Lipman et al. (2023) introduced Flow Matching (FM), a simulation-free approach for training Continuous Normalizing Flows. FM models, when optimized with Optimal Transport learning objective (OTFM), have demonstrated improved sample quality compared to diffusion-based methods in the domain of 2D natural images (Lipman et al., 2023; Esser et al., 2024). However, whether the improvements observed with OTFM in 2D natural images translate to 3D medical image generation remains an open question, as most recent works in this domain rely on diffusion-based approaches, which currently dominate the field (Pinaya et al., 2022; Khader et al., 2023; Friedrich et al., 2024; Wang et al., 2025).

In this work, we investigate the use of Flow Matching in the 3D medical imaging domain, more specifically in the context of 3D craniofacial skeletal data generation. We use our trained model to generate synthetic samples and show via quantitative and qualitative analysis how they effectively capture the main anatomical structures. We compare synthetic datasets generated using OTFM and DDPM, finding that OTFM surpasses DDPM in generating more realistic 3D data. Moreover, we show how OTFM leads to a more robust generation, i.e. a lesser number of samples with unplausible anatomical structures. Finally, we test our best synthetic dataset in two clinical downstream tasks, namely skull alignment and shape completion, assessing its utility in both augmenting and substituting real data. We also demonstrate that synthetic data can be used to balance datasets to improve model performance.

## 2. Related work

In recent years, image generation has been advancing rapidly, evolving from GANs (Heusel et al., 2017), which produce high quality images but lack diversity and can suffer from mode collapse (Che et al., 2017), to DDPMs (Ho et al., 2020), which generate more diverse images but at the cost of lower resolution. Latent Diffusion Models address these limitations by enabling high-resolution image generation exploiting latent spaces, therefore improving efficiency and scalability (Rombach et al., 2022). Recently, Flow Matching emerged as a novel generative modelling paradigm that subsumes DDPMs and enables more robust and

stable training compared to diffusion models (Lipman et al., 2023). FM models learn to map noise to data by matching probability flow trajectories between two distributions. A particular instance of FM, Optimal Transport Flow Matching, implements this mapping as a straight line between samples drawn from the two distributions. In the domain of 2D natural image generation, this approach outperforms diffusion models in terms of both likelihood and sample quality (Lipman et al., 2023; Esser et al., 2024). However, OTFM has not yet been extensively studied in other domains, such as 3D medical image generation.

The medical domain poses its own challenges related both to the 3D nature of most medical imaging techniques, such as CTs and MRIs, and to the sensitivity of clinical applications, which require anatomically plausible synthetic data to avoid introducing harmful biases. The generation of 3D medical data has closely followed the advances in 2D natural image generation. GANs have been widely used for 3D image generation across different modalities and anatomical regions: to generate 3D Time-of-Flight Magnetic Resonance Angiography patches, brain MRI, thorax CTs, liver and spine CTs (Subramaniam et al., 2022; Sun et al., 2022; Kim et al., 2024). However, due to their unstable training that often results in mode collapse, GANs have increasingly been replaced by diffusion-based models. Pinaya et al. (2022) leveraged LDMs to generate synthetic data from high-resolution 3D brain MRIs. Khader et al. (2023) proposed a similar approach using four different datasets with about 1000 elements each, showing that it is possible to train generators with datasets of limited size. Friedrich et al. (2024) applied diffusion on wavelet decomposed 3D images for improving efficiency. Wang et al. (2025) presented a patch-wise autoencoder and a novel denoiser to generate both MRIs and CT scans. Recently, Yazdani et al. (2025) tested OTFM with 2D echocardiographic images and 3D MRI data, but the study related to 3D data addresses only low resolution volumes (138x169x138) and quantitative evaluation is limited to generation quality metrics, which may not necessarily transfer to utility in clinical downstream tasks. Conversely, we rely on OTFM to train a generative model with high resolution 3D craniofacial skeletal data (456x352x512). We compare its performance with that of a diffusion-based approach to assess whether the improvements observed in general 2D image generation extend to 3D medical image generation. Furthermore, we validate the generated synthetic data by evaluating their impact on two clinical downstream tasks.

## 3. Method

### 3.1. Flow Matching

We start with a brief review of Flow Matching, a generative modelling technique introduced by Lipman et al. (2023). The aim of FM is to learn a mapping between two distributions, a complex distribution $P_0$, the one of the real data, and a simple distribution $P_1$, usually modelled as a standard normal distribution $\mathcal{N}(0, 1)$. Similarly to DDPMs, core components of FM are the forward and backward processes. The forward process is a function describing the noising process of a data sample. Given a sample $x_0$ from $P_0$ and a timestep $t$, it returns a noised version of the sample $z_t$. It can be seen as a flow of the sample between $P_0$ and $P_1$:

$$z_t = a_t x_0 + b_t \epsilon \quad \text{where} \quad \epsilon \sim \mathcal{N}(0, 1) \tag{1}$$

$t$ is the timestep describing how far in the flow we are, and it is uniformly distributed between 0 and 1. $a_t$ and $b_t$ are two functions describing the noising process. When $t = 0$,

$a = 1$ and $b = 0$, therefore $z = x_0$, i.e. we did not move in the flow. Conversely, when $t = 1$ we have that $a = 0$ and $b = 1$, therefore $z = \epsilon$, i.e. the flow of any data sample ends into a realization of standard Gaussian noise. The flow is modelled by a vector field $u_t$, which returns the direction of the flow for a specific point. It is defined as the derivative of the flow $u_t = \frac{dz_t}{dt}$. By learning the vector field $u_t$ with a neural network parametrized by $\Theta$, we learn the mapping between the two distribution $P_0$ ad $P_1$ and, by traversing the flow, we are able to generate new data from the distribution $P_0$. To learn the vector field Lipman et al. (2023) proposed a tractable learning objective called Conditional Flow Matching (CFM), which allows us to supervise the learning using pair of samples $x_0 \sim P_0$, $\epsilon \sim P_1$, and a randomly sampled timestep $t \in [0, 1]$:

$$\mathcal{L}_{CFM} = \mathbb{E}_{t, \, p_t(z|\epsilon) \, p(\epsilon)} \|v_\Theta(z, t) - u_t(z|\epsilon)\|_2^2 \tag{2}$$

Different ways of defining the forward process, i.e. different implementation of $a_t$ and $b_t$, define different flows between the two distributions $P_0$ and $P_1$. An interesting case is the one that recovers DDPMs learning objective (Lipman et al., 2023; Esser et al., 2024), which means that FM subsumes DDPMs. A more intuitive way to define the mapping between the distributions is to use Optimal Transport, i.e. a straight path between them:

$$a_t = 1 - t, \quad b_t = t \tag{3}$$

This means that in Flow Matching with Optimal Transport (OTFM) (Lipman et al., 2023) the forward process for a single sample, and the ground truth vector field that the network aims to learn, can be described as:

$$z_t = (1 - t)x_0 + t\epsilon, \qquad u_t = \epsilon - x_0 \tag{4}$$

In practice, at training time, for a given image we sample a timestep $t \in [0, 1]$ and a noise sample $\epsilon \sim \mathcal{N}(0, 1)$. We compute the noised version of the image $z$ as per Equation (4) and we pass $t$ and $z$ as input to a denoiser network, which computes the vector field $v_\Theta(z, t)$. The loss is then the difference between the predicted vector field and the ground truth one. Conversely, at inference, given a number of steps $S$, we start from standard Gaussian noise $z = \epsilon$ at $t = 1$ and predict the vector field $v_\Theta(z, t)$ with the trained network. We compute the updated sample and timestep by moving in the predicted direction in a small step $z_{new} = z - \frac{v_\Theta(z,t)}{S}$ and $t_{new} = t - \frac{1}{S}$. We perform this operation $S$ times to reach a synthetic sample from the data distribution $P_0$.

### 3.2. Architecture, training and inference

We use OTFM to train a generator to synthesize novel 3D skeletal data, but to generate volumes at a reasonable resolution it is not feasible to work directly in the data space, as it would require prohibitive memory resources. Instead, we apply the generative process in a lower-dimensional latent space. As illustrated in Figure 1, to enable this we design two main components starting from the work of Khader et al. (2023): an autoencoder, trained in the first stage; and a denoiser, which is trained in the second stage to generate novel but realistic samples within the latent space.

Training the autoencoder with the data at their highest resolution is often prohibitive in terms of memory consumption, forcing volumes to be downsampled during preprocessing

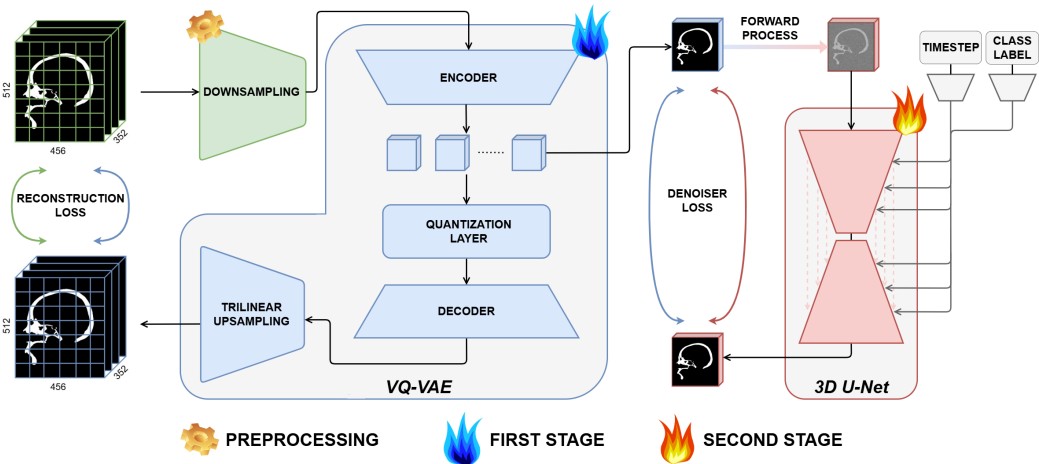

Figure 1: Architecture of the proposed generative model. The model is trained in two stages: the VQ-VAE training, with the loss being computed at the highest available resolution; the denoiser training, with OTFM or DDPM learning objective depending on the experiment.

and causing information loss. We designed the autoencoder as a VQ-VAE (Razavi et al., 2019) modified to minimize the information loss caused by the downsampling. The VQ-VAE, in blue in Figure 1, is made of three main components: an encoder, a decoder, and a vector quantization layer. The latter consists of a codebook of $n$ vectors whereby each element of the dense representation is replaced by the nearest code vector (in Euclidean distance), resulting in a discrete latent representation of the input data. To retain high resolution information, as shown in Figure 1, we added a deterministic trilinear upsampling layer as last layer of the decoder. In this way, the reconstruction loss of the autoencoder is computed at the highest resolution possible while causing negligible memory consumption. The VQ-VAE is optimized with two loss components: a reconstruction loss $L_{rec}$ defined as the L1 distance between a volume $x$ and its reconstruction $\hat{x}$; and a commitment loss $L_{commit}$, the mean squared error between the encoder's output and the selected code vector. Each vector of the codebook is optimized by maintaining an exponential moving average of all the dense vectors that get mapped to it.

The denoiser, in red in Figure 1, is implemented as a 3D U-Net (Ho et al., 2022). The model is optimized with OTFM, but we also train a variant with the DDPM learning objective to conduct a comparative experimental evaluation (see Section 4). The training pipeline is identical across all experiments independently of the objective. The model receives as input the latents of the training volumes, normalized to be approximately in the $[-1, 1]$ range. Following Khader et al. (2023), this is done via min-max normalization using as bounds the minimum and maximum values that can be found in the learned codebook. According to the forward process, the noised latent $z_t$ (with $t$ being sampled from a discrete uniform distribution defined over 300 timesteps) is computed and successively fed to the network to regress the vector field, as described in Section 3.1. In addition, during training, the model is conditioned on a class label, which is embedded and concatenated with the

timestep embedding to produce a conditioning vector. This conditioning vector is used to scale and shift the intermediate activations of the convolutional blocks of the U-Net. In our setting, the generation is conditioned on the Quality Score, which is used as the class label and is a feature of the data described in Section 3.3.

At inference time, new volumes are generated by reversing the forward process. Starting from $z_t$ sampled from the distribution $P_1$, i.e. pure noise and $t = 1$, we iteratively query the denoiser and use the output to compute the $z_{t-1}$, until we reach $t = 0$. We do this in $S = 300$ steps. The generated latent is then quantized and decoded to yield a synthetic sample. The resulting volume is a voxel grid with the same size as the original dataset, and it can be further processed to obtain a mesh via the marching cube algorithm (Lorensen and Cline, 1987). A more detailed description of our architecture, together with the training hyperparameters, is provided in the supplementary material, Section B, while the ablation studies on the architectural choices are presented in Section C of the supplementary.

### 3.3. Dataset and preprocessing

The data used in this work is the union of two different datasets of anonymized head CT scans: CQ500, a publicly available dataset of 355 scans introduced by Chilamkurthy et al. (2018); and a private dataset of 591 scans from Bologna's Sant'Orsola hospital[1]. An expert segmented the volumes to obtain meshes depicting the skeletal part of the head, and subsequently aligned them to a reference skull. Since the CTs have been acquired for different purposes, most of the scans do not depict the entirety of the skull. Therefore, the same expert labelled them with a Quality Score (QS), which describes the extension and completeness of each shape: as illustrated in Figure 2(a), QS 1 data represent the least complete skulls; QS 2 data include a complete skull cap but only a very limited portion of the nasal area; QS 3 data also contain a complete skull cap, along with a larger portion of the nasal area; QS 4 data may lack the skull cap but must contain the complete nasal area; QS 5 data contain almost complete skulls while QS 6 scans contain the mandible but miss the upper part of the skull. Quality Score 1 meshes are not informative enough to be kept for training the models and have been removed from the dataset. Moreover, other scans had been manually labelled as not suitable and thus have been discarded. The final dataset is made of 908 scans, 341 from the public dataset and 567 from Sant'Orsola's dataset, and is split in a stratified fashion to obtain a train (726) and a test set (182). The distribution of Quality Scores among the splits is depicted in Figure 2(b), which shows that the dataset is imbalanced, as there are fewer QS 6 volumes than all other Quality Scores. We will leverage conditioned generation to mitigate the imbalance and evaluate its effectiveness in clinical downstream tasks.

Since the models require voxels as input data, the first data preprocessing step consists in the voxelization of the meshes, which is done using an isotropic spacing of $0.51mm$, resulting in cubic voxel grids of side 512. To reduce memory consumption, the grids are subsequently cropped to get rid of regions that are empty across all training skulls, leading to grids of size 456x352x512. As described in Section 3.2 and illustrated in Figure 1, this represents the size at which the autoencoder loss is computed thanks to the upscaling performed by

---

1. The use of this dataset for research purposes was approved by the local institution. Protocol details will be provided upon acceptance.

the trilinear upsampling component, i.e. the output size of the model. On the other hand, as shown in the downsampling block of Figure 1, due to computational constraints, the input volumes are downsampled with a scaling factor $s = 0.82$, leading to input grids of size 374x289x420. Finally, the volumes are min-max normalized to the range $[-1, 1]$.

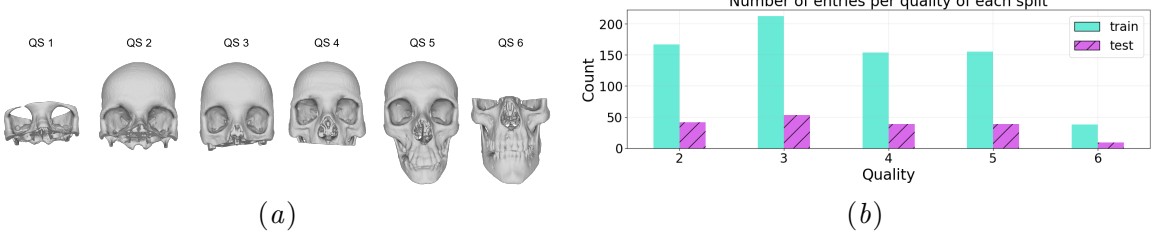

Figure 2: (a): Examples of meshes for each Quality Score. (b): Distribution of the Quality Scores in the train and test splits.

## 4. Experiments and discussion

### 4.1. Evaluation protocol

The trained models are used to generate synthetic datasets that are evaluated both qualitatively and quantitatively to assess their quality. For the qualitative evaluation, we provide both central slices of synthetic voxel grids and the generated meshes. Whereas for the quantitative evaluation, multiple metrics are computed to evaluate different aspects of the generation quality. More in detail, the Fréchet Inception Distance (FID) (Heusel et al., 2017) is used to measure within a single value the quality of the generated dataset by comparing the feature distributions of real and synthetic data. This metric has already been used in different works concerning 3D medical data generation (Pinaya et al., 2022; Wang et al., 2025; Yazdani et al., 2025; Kim et al., 2024; Friedrich et al., 2024). Since the FID is a single scalar value, it does not allow us to disentangle realism from diversity in the generations. To do so, we compare real and synthetic data using precision and recall, with the first measuring the ability of the model of generating realistic volumes and the second indicating how much the generated dataset covers the whole spectrum of real data. There exist different methods to compute precision and recall, and in this work we used Improved Precision and Recall (IP, IR) (Kynkäänniemi et al., 2019), as well as Probabilistic Precision and Recall (PP, PR) (Park and Kim, 2023).

All the metrics described above (e.g. FID, precision, and recall) are computed in the feature space of a pretrained network. In particular, similarly to other works related to medical data (Pinaya et al., 2022; Friedrich et al., 2024; Wang et al., 2025), we exploit the features extracted using Med3D (Chen et al., 2019), a 3D convolutional network trained on aggregated datasets from different medical tasks. The features are extracted by applying Global Average Pooling to the final layer before classification, resulting in feature vectors of size 2048.

Moreover, we use Multi-Scale Structural Similarity Index Measure (MS-SSIM) (Wang et al., 2003) to evaluate the diversity within a dataset by computing the similarity between 1000 random pairs of volumes of the same set and averaging the results. The lower the measure, the more diverse a dataset is.

## 4.2. Generation results

Using the architecture presented in Section 3.2, we trained generative models both with DDPM and OTFM learning objectives. These models were used to synthesize datasets with the same cardinality and QS distribution as the original training set. Table 1 shows the quantitative results on the synthetic datasets. To provide a reference for these metrics on real data, under the *ref* setting we computed FID, precision, and recall between the training and test sets of the real data, and MS-SSIM on the real training set. Overall, the results indicate that the OTFM model generates higher quality data, as the FID is lower with respect to DDPM. While in term of realism OTFM beats DDPM on all metrics, the results on diversity are more metric dependent, with DDPM obtaining better results in MS-SSIM and PR. Our hypothesis is that this result is influenced by the presence of anatomically unplausible generations, which would improve diversity at the expense of realism. The qualitative analysis of the generated volumes (Figure 3) shows that the OTFM model is more robust, leading to spatially consistent and realistic slices. Moreover, it confirms that the data generated with DDPM do not always represent the Quality Score with which they were conditioned and are sometimes not realistic and completely out of distribution. This is coherent with the quantitative analysis, as wrongly generated data could increase the diversity of the dataset while lowering the realism. Since both models use the same autoencoder model, the results are directly connected to the adopted denoising process. The generated meshes depicted in Figure 4(a) show that the main anatomical structures are built correctly, even for smaller details such as the cranial nerve holes visible in the QS 6 mesh or the ear canal visible in QS 2 and QS 3 ones. The resolution is high enough to reach a discrete smoothness of the meshes, even if it is not at the level of the real ones.

Table 1: **Quantitative evaluation of the generation quality.** Under the *ref* setting, FID, precision, and recall are computed comparing the train and test sets of the real data, while MS-SSIM is calculated on the real training set.

|      | FID ↓   | MS-SSIM ↓ | IP ↑  | IR ↑  | PP ↑  | PR ↑  |
| ---- | ------- | --------- | ----- | ----- | ----- | ----- |
| ref  | 0.0012  | 0.59      | 0.93  | 0.92  | 0.8   | 0.76  |
| DDPM | 0.073   | **0.59**  | 0.55  | 0.73  | 0.27  | **0.76** |
| OTFM | **0.0024** | 0.63   | **0.96** | **0.82** | **0.87** | 0.58  |

## 4.3. Downstream tasks

To further evaluate the generation quality we assess the performance of the synthetic datasets in two clinical downstream tasks: shape completion and skull alignment. These

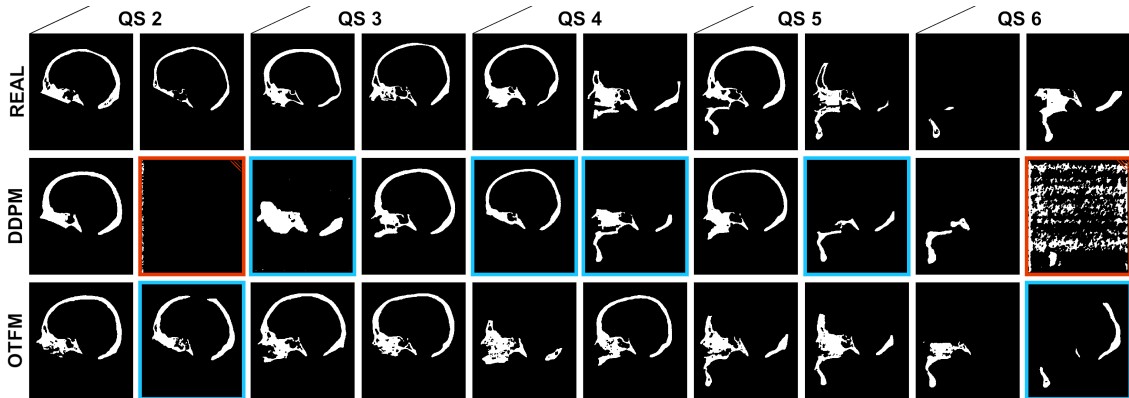

Figure 3: Middle slice of the sagittal view of head CT scans. The first row shows real scans, the second shows volumes generated with DDPM, the third shows generations using OTFM. The generated scans have been chosen randomly from the respective datasets. Red borders highlight failed generations, while blue borders point out scans that do not depict the right QS, the one they were conditioned with.

tasks are part of pre-surgical operations performed in maxillofacial surgery and are usually performed manually by an expert. The current method of treating malformations of the craniofacial skeleton requires dedicated technicians to manipulate 3D medical images, simulating the correction of the defect targeted by the surgery. In this context, shape completion consists in automatically reconstructing defected skulls to speed up surgical planning and providing patient-specific hints (Mazzocchetti et al., 2024). To train this model all data must be aligned to a canonical reference, which is not the case in real clinical data due to differences in acquisition devices and patient positioning. For this reason, to use the shape completion tool in a clinical setting, we need another model trained to align skulls.

Since data availability in the medical domain is often an issue, we decided to conduct the experiments in different scenarios, following the protocol adopted in Wang et al. (2025) to create the training sets: **100% real** - there is no synthetic data, the model is trained with the original training set (726 samples); **100% synth** - there is no real data available, the model is trained with all the samples from the synthetic dataset (726 samples); **100% real + 25% synth** - an augmentation setting where we add to the original training set 25% of the synthetic data samples ($726 + 182 = 908$ samples); **50% real** - there is no synthetic data, the model is trained with half of the samples of the original training set to simulate data scarcity (363 samples); **50% real + 25% synth** - the new training set is made of half of the real training set and 25% of the synthetic data samples to simulate augmentation in a data scarcity setting ($363 + 182 = 545$ samples); With these experiments, we aim at measuring whether synthetic data can be used in place of real data and whether they are effective in augmenting existing datasets. Moreover, we designed another experiment to evaluate the possibility of using synthetic data to balance datasets. To do this, we used the model trained with OTFM to augment the real dataset so as to obtain a balanced one, leading to a total cardinality of 1060 volumes, 212 volumes for each Quality Score. To make

a fair comparison, we also augmented the real dataset in a stratified fashion to obtain the same cardinality as the one in the balanced set, but keeping the QS distribution unbalanced.

**Aligner.** The first task consists in aligning segmented CT scans to a canonical reference. The neural network that learns the alignment is built on top of PointNet++ (Qi et al., 2017) and it is described in detail in the supplementary, Section E.1. The model takes as input skulls represented as point clouds. Since the training set is already aligned, we adopt a self-supervised training strategy by applying random roto-translations to the input point clouds on-the-fly during training. The aligner is trained to regress the roto-translation matrix that recovers the ground-truth aligned position. The trained model is tested on a set of roto-translated skulls and evaluated with the average per point distance.

The quantitative results in Table 2 show that OTFM-generated data are more effective than DDPM-generated data for this clinical downstream task, as OTFM outperforms DDPM in all settings. When augmenting real data (Table 3), OTFM leads to substantial performance gains, especially in the setting that simulates data scarcity. Analysing the performance across each Quality Score, we noticed that the model is particularly sensitive to the QS of the skulls, performing poorly with QS 6 skulls, i.e. the least represented Quality Score. The results in Table 4 show that balancing the dataset is very effective in solving this issue. The network, receiving many more QS 6 volumes, learns to align them more precisely leading to a better performance in that class. Moreover, the overall score improves, meaning that a balanced dataset helps the model to generalize and is not detrimental for previously over-represented classes.

Table 2: **Aligner evaluation: OTFM vs DDPM.** It shows the average per point distance in $mm$ between roto-translated and ground truth skulls.

|  | 100% synth | 100% real + 25% synth | 50% real + 25% synth |
|---|---|---|---|
| DDPM | 8.11 | 6.93 | 9.73 |
| OTFM | **7.31** | **6.48** | **9.17** |

Table 3: **Aligner evaluation: OTFM vs real data.** It shows the average per point distance in $mm$ between roto-translated and ground truth skulls. *real only* denotes that only real data have been used to train the skull alignment model.

|  | 100% real | 50% real |
|---|---|---|
| real only | 6.95 | 30.8 |
| +25% OTFM | **6.48** | **9.17** |

**Shape completion.** To automatically reconstruct malformed skulls we followed the work of Wang et al. (2024) and used PointAttN, a network that exploits attention to construct a point cloud completion network. In order to generate the defects we replicated the pipeline of Mazzocchetti et al. (2024), which consists in the removal from the input point cloud of cuboids sized between $3cm$ and $10cm$. The partial point clouds used for

Table 4: **Aligner dataset balancing experiment.** Average per-point distance in *mm* between roto-translated and ground truth skulls. *OTFM-s* is the real dataset augmented in a stratified fashion. *OTFM-b* is the real dataset augmented to balance the classes cardinality.

|           | overall | QS 2 | QS 3 | QS 4 | QS 5 | QS 6  |
|-----------|---------|------|------|------|------|-------|
| 100% real | 6.95    | 6.47 | 4.85 | 6.53 | 7.08 | 21.6  |
| OTFM-s    | 5.87    | **5.34** | 4.72 | **5.41** | **5.22** | 17.87 |
| OTFM-b    | **5.67** | 5.54 | **4.60** | 5.71 | 5.28 | **8.12** |

training are generated online, while validation and test clouds are kept fixed throughout all the experiments. To evaluate the model we used Chamfer Distance (CD), Accuracy and Completeness, which were computed considering just the defective region.

The results reported in Table 5 and Table 6 highlight that the models trained with OTFM-generated data outperform the ones trained with DDPM-generated data, and that augmenting the real dataset improves performance. Since we found experimentally that the performance of the model does not vary greatly for different QSs, balancing the dataset is less critical in this task. Therefore, the results on the balancing experiment are reported in the supplementary, Section E.3.

Table 5: **Shape completion evaluation: OTFM vs DDPM.** For each experimental setting we show Chamfer Distance (CD), Accuracy and Completeness in millimeters.

|      | 100% synth | | | 100% real + 25% synth | | | 50% real + 25% synth | | |
|------|---------|----------|-----------|---------|----------|-----------|---------|----------|-----------|
|      | CD ↓ | Acc ↓ | Comp ↓ | CD ↓ | Acc ↓ | Comp ↓ | CD ↓ | Acc ↓ | Comp ↓ |
| DDPM | 3.50 | 4.34 | 2.66 | 3.30 | 4.04 | 2.56 | 3.37 | 4.15 | **2.60** |
| OTFM | **3.36** | **4.09** | **2.63** | **3.17** | **3.81** | **2.54** | **3.31** | **3.99** | 2.63 |

Table 6: **Shape completion evaluation: OTFM vs real data.** For each experimental setting we show Chamfer Distance (CD), Accuracy and Completeness in millimeters. *real only* denotes that only real data have been used to train the shape completion model.

|           | 100% real | | | 50% real | | |
|-----------|---------|----------|-----------|---------|----------|-----------|
|           | CD ↓ | Acc ↓ | Comp ↓ | CD ↓ | Acc ↓ | Comp ↓ |
| real only | 3.29 | 3.98 | 2.6 | 3.41 | 4.12 | 2.69 |
| +25% OTFM | **3.17** | **3.81** | **2.54** | **3.31** | **3.99** | **2.63** |

### 4.4. Failure cases

There are few cases in which OTFM fails at generating accurate data. By qualitatively analysing these cases, we can define two main sources of defects: those connected to the nature and quality of the training data, and those related to the generation process. An example mesh of the first type of error, showing anatomically unplausible holes in the skeletal structure, is depicted in the left side of Figure 4(b). As explained in Section 3.3, the thresholding applied to obtain the skeletal component of a CT scan can lead to holes in the resulting skull. This means that the training set data will contain examples with non-anatomical holes as well. As a result, the model will learn to imitate this feature, even if it is an unwanted one. As regards errors emerging from the generative process, a second example reported in the right side of Figure 4(b) shows defects connected to the generation of high-frequency details. More in detail, the generated QS 5 mesh contains dental arches that are not sufficiently sharp, resulting in smoother volumes with less anatomical precision.

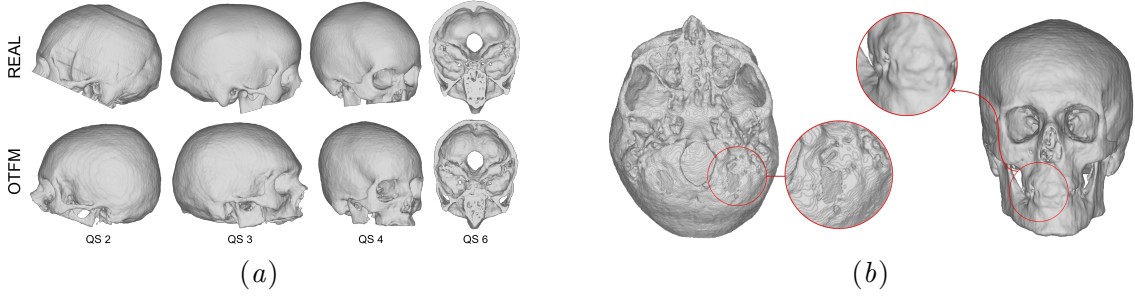

Figure 4: (a): Real and generated meshes. (b): Generated meshes with some failures.

## 5. Conclusion

In this work, we investigated the use of Flow Matching with Optimal Transport to generate 3D craniofacial skeletal data. Despite the challenges posed by the three-dimensional nature of medical data and the high level of clinical accuracy required, we showed that this generative modelling paradigm succeeds in generating high quality synthetic data. In addition to the quantitative and qualitative analysis, we validated the results on two medical downstream tasks, confirming the validity of the data generated with our solution for various settings. The robust generation broadens the spectrum of available tools in 3D medical data synthesis, and the performance in the downstream tasks confirms their relevance in augmenting real datasets, in replacing them when they are not available and in balancing them, which can be critical for various tasks. Moreover, the synthetic dataset obtained with OTFM was compared to data generated with DDPMs, the dominant paradigm in 3D medical images generation. The results showed how OTFM surpasses DDPM in terms of generation quality, robustness of the model, and ability in conditioning the generation with Quality Scores. Despite this, there are still a few cases in which OTFM generations contain small errors. Therefore, future work will focus on injecting anatomical knowledge into the model and enhancing the generation of high-frequency details of the volumes.

## Acknowledgments

We acknowledge and thank Gabriele Cialone for his work on the aligner architecture used for the skull alignment downstream task. We acknowledge the CINECA award under the ISCRA initiative, for the availability of high-performance computing resources and support.

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

## Appendix A. Code availability

The code is available at the following link:
https://github.com/Chavelanda/skeletal_fm.

## Appendix B. LDM architecture and hyperparameters

This section provides a detailed description of the architecture used for generating synthetic volumes introduced in Section 3.2, together with the hyperparameters used for training and inference. Both the autoencoder and the denoiser have been trained on 4 NVIDIA Ampere A100 with $64GiB$ of GPU RAM for approximately 5 days.

### B.1. VQ-VAE

The architecture used to learn a compressed representation of the volumes is a VQ-VAE (Esser et al., 2021) modified starting from the version of Khader et al. (2023) to compute the loss at the highest available resolution. Given a dataset $D$ containing volumes $v^* \in \mathbb{R}^{1\mathrm{x}D\mathrm{x}H\mathrm{x}W}$, the encoder of the VQ-VAE is fed with pre-processed downsampled volumes $v \in \mathbb{R}^{1\mathrm{x}d\mathrm{x}h\mathrm{x}w}$ to compute dense latent representations $z \in \mathbb{R}^{k\mathrm{x}\frac{d}{s}\mathrm{x}\frac{h}{s}\mathrm{x}\frac{w}{s}}$, where $d$ is the depth, $h$ is the height, and $w$ is the width of the pre-processed volume, $k$ is the number of channels in the latent and $s$ is the encoder compression factor. In the quantization step, each latent feature vector $z_i \in \mathbb{R}^k$ is replaced by the closest code $q_i = Q(z_i)$ contained in the codebook $Q$. The decoder, receiving the quantized latent $q$, is used to reconstruct the volume $\hat{v} \in \mathbb{R}^{1\mathrm{x}d\mathrm{x}h\mathrm{x}w}$, which is then upsampled with trilinear upsampling to compute $\hat{v}^* \in \mathbb{R}^{1\mathrm{x}D\mathrm{x}H\mathrm{x}W}$. The autoencoder is trained with two different losses: a reconstruction loss $L_{recon} = ||v^* - \hat{v}^*||_1$ and a commitment loss $L_{commit} = \frac{1}{I}\sum_{i=0}^{I}||z_i - q_i||_2^2$, where $I = \frac{d}{s} * \frac{h}{s} * \frac{w}{s}$. The encoder is a sequence of blocks that first downsample the input via a 3D convolution and then process it through a residual block. The decoder is similar, with each block having a strided 3D convolution to upsample the input followed by two residual blocks. Table 7 shows the hyperparameters for training the model.

### B.2. 3D U-Net

The 3D U-Net (Ho et al., 2022) is the network used to denoise the latents. It follows the traditional encoder-bottleneck-decoder structure, and is conditioned on both the noising timestep $t$ and the class $c$, i.e. the Quality Score. The timestep $t \in [0, T[$ is embedded using a sinusoidal positional embedding followed by two linear layers with a GELU activation in between. The class $c$ is embedded using a learned embedding layer to match the dimensionality of the timestep embedding. The embeddings of $t$ and $c$ are concatenated together to form the conditioning vector. To adapt the UNet architecture to volumetric data, each 2D convolution is substituted with a 3D convolution using 3x3x3 kernels. Conditioning is incorporated in the convolutional blocks by scaling and shifting the intermediate activation with the conditioning vector. Each convolutional block is followed by two attention layers: first, a 2D spatial attention over $H$ and $W$, where $D$ is treated as batch axis; secondly, a 1D attention across the depth dimension $D$, with all other dimensions treated as batch axes. The details on the hyperparameters used for training the model are shown in Table 7.

Table 7: **Hyperparameters for training the VQ-VAE and the 3D U-Net.**

| VQ-VAE | |
|---|---|
| Encoder compression factor $s$ | 4 |
| Codebook size | 16384 |
| Codebook dimensionality $k$ | 8 |
| Batch size | 1 |
| Learning rate | 1e-4 |
| Num. training iterations | 150 000 |
| **3D U-Net** | |
| Batch size | 1 |
| Learning rate | 5e-5 |
| No. training iterations | 200 000 |
| Timesteps T | 300 |

## Appendix C. Autoencoder ablation studies

In this section, we validate different aspects of the autoencoder: the learning objective, the architecture and the effect of the input volume resolution.

### C.1. Learning objective

Khader et al. (2023), the work we started from to develop our architecture, employed a VQ-GAN to build the latent space, which is a modified VQ-VAE trained with both the reconstruction loss and an adversarial component (Esser et al., 2021). Moreover, the original work included also a perceptual LPIPS loss (Zhang et al., 2018) computed on random slices of the volumes. We run experiments with both the VQ-VAE and VQ-GAN formulations to evaluate which one provided better reconstructions of craniofacial skeletal data, and we conducted an ablation on the perceptual loss component to assess its utility. The VQ-VAE training was faster and reached lower losses with respect to the VQ-GAN one, and the results in Table 8 show that the reconstruction is better when using the simpler loss described in Section 3.2, without the adversarial component and without the perceptual loss.

### C.2. Architecture and input resolution.

In the 3D data generation pipeline, the autoencoder reconstruction quality acts as an upper bound for the quality of generated data. The quality is also constrained by the resolution of the input training volumes, which is often limited due to memory reasons. This is why we decided to investigate the difference in the reconstruction quality with or without the final upsampling layer and compared the models trained with different input volume resolutions. To evaluate the performance of the autoencoder we use the reconstruction error, which is always computed with respect to the highest resolution input volume. Lower resolution volumes are upsampled during post-processing to make the computation possible. The settings we tested are the following. $s=0.422$, **up**: The downsampling factor is 0.422

and there is the final upsampling layer, i.e. the loss is computed at the highest resolution. **s=0.53, up**: the downsampling factor is 0.53 and there is the final upsampling layer. **s=0.578, no up**: the downsampling factor is 0.578 and there is no final upsampling layer, i.e. the loss is computed at the downsampled resolution. **s=0.578, up**: the downsampling factor is 0.578 and there is the final upsampling layer. **s=0.82, up**: the downsampling factor is 0.82 and there is the final upsampling layer.

By comparing the results on experiments with same downsampling factor ($s = 0.578$) but different architecture, we can observe that upsampling the volume before loss computation is beneficial for the reconstruction quality. As expected, the input volume resolution correlates with reconstruction quality, but the quality gain decreases as the resolution goes higher. The downsampling factor we used in our main experiments, $s = 0.82$, is the one that yields the smallest resolution loss given the memory constraints.

Table 8: **Autoencoder ablation results.** Left: reconstruction error for different autoencoders. Right: reconstruction error for different input volume resolutions and different autoencoder architectures.

| Autoencoder | Reconstruction error ↓ |
|---|---|
| VQ-GAN | 0.02323 |
| VQ-VAE + LPIPS | 0.00293 |
| VQ-VAE | **0.00227** |

| Res, Model | Reconstruction error ↓ |
|---|---|
| s=0.422, up | 0.0037 |
| s=0.53, up | 0.0033 |
| s=0.578, no up | 0.0039 |
| s=0.578, up | 0.0025 |
| s=0.82, up | **0.0023** |

## Appendix D. Generation results supplementary

In this section, we present the results of an analysis conducted on synthetic datasets generated by the OTFM-based and DDPM-based models, aimed at quantifying the occurrence of failed generations and conditioning errors in both cases. As highlighted by Table 9, OTFM demonstrates greater robustness in generating novel samples, resulting in fewer completely out-of-distribution samples and more accurate conditioning.

Table 9: **Failure analysis experiment.** Percentage of failed generations and conditioning errors for DDPM-based and OTFM-based models.

| | failed generations | conditioning errors |
|---|---|---|
| DDPM | 12% | 32% |
| OTFM | **4%** | **6%** |

## Appendix E. Downstream tasks supplementary

This section provides further explanations and experiments on the downstream tasks. We describe the aligner neural network, and provide the hyperparameters to train it. Also the details to train the shape completion network are provided, together with the results on the dataset balancing experiment.

### E.1. Aligner architecture

In this section we describe the architecture of the aligner network. The network is made of two main components: firstly, one that estimates the rotation of the volume; secondly, one that refines the estimate and regresses a translation vector. As explained in Section 4.3, the model takes as input a roto-translated point cloud. The cloud is encoded to obtain an high dimensional embedding using a modified PointNet++ (MSG), one of the modules of PointNet++ (Qi et al., 2017). PointNet++ (MSG) is modified by adding a skip connection from the output of the first Set Abstraction layer to the output of the encoder. The resulting embedding vector is processed by a rotation head, which is made of two MLP blocks followed by a linear layer that regresses a flattened rotation matrix. Each MLP block reduces the vector dimensionality by four and is made of a linear layer, followed by batch normalization and ReLU activation function. The rotation matrix regressed by the rotation head is used to obtain an approximate rotated skull.

The second component is similar to the first one, and processes the approximate skull to obtain a finer rotation and a translation vector. The rotated skull is encoded with the modified PointNet++ (MSG), and the embedding is fed to the rotation head to recover another, finer, rotation matrix. Additionally, the embedded representation of the point cloud is also fed to a translation head to retrieve a translation vector. The translation head is similar to the rotation one: it is made of two MLP blocks followed by a linear layer, but in this case the final linear layer regresses a translation vector. The final rotation matrix and translation vector are used to compute the rotated skull. The L1 loss between the predicted skull and the ground truth one is calculated and used to train the network.

### E.2. Networks hyperparameters

Details on the hyperparameters of both the aligner and the shape completion networks are provided in Table 10.

### E.3. Shape completion balancing experiment

The results of the experiment on balancing the training set in the shape completion downstream task (Section 4.3) have not been reported in the main paper because the baseline results, i.e. the results obtained using only real data, did not show an impact of the Quality Score in the performance of the model. As it is possible to observe from Table 11, the average Chamfer Distance between the predicted and ground truth point clouds does not correlate with the cardinality of the QS. On the contrary, the model performs better with QS 6 data. This is a task for which it would not make sense to use use the generator to create a balanced dataset. Still, for completeness, we report the results of the balancing experiment.

Table 10: **Hyperparameters of downstream tasks networks.**

| Aligner | |
|---|---|
| No. training iterations | 7000 |
| Loss | L1 |
| Num. input points | 8192 |
| Batch size | 12 |
| Learning rate | 0.001 |
| Embedding dim | 1024 |
| **Shape completion** | |
| No. training iterations | 72800 |
| Loss | Chamfer Distance |
| Num. input points | 4096 |
| Batch size | 4 |
| Learning rate | 0.0001 |

Table 11: **Shape completion experiment on dataset balancing using OTFM.** *otfm-s* is the real dataset augmented in a stratified fashion. *otfm-b* is the real dataset augmented to balance the classes cardinality.

| | overall | QS 2 | QS 3 | QS 4 | QS 5 | QS 6 |
|---|---|---|---|---|---|---|
| 100% real | 3.29 | 3.26 | 3.09 | 3.89 | 3.09 | 3.02 |
| OTFM-s | 3.17 | 3.16 | 3.02 | **3.52** | 3.12 | **2.96** |
| OTFM-b | **3.14** | **3.06** | **2.99** | 3.61 | **2.99** | 3 |

Table 11 shows that the distribution of the Quality Scores in the training set is not significant for model performance. The model trained with the balanced dataset and the model trained with the augmented stratified dataset have very similar overall performance. Moreover, the performance on each QS does not correlate with the cardinality of the class, confirming that in this task balancing the dataset is not useful.

