# OpenReview forum: "Flow Matching for 3D Craniofacial Skeletal Data Generation"
_MIDL.io/2026/Conference — MIDL 2026 Poster_

### Official Review · Reviewer_PqiT · 2025-12-29

**Confidence:** 4
**Preliminary Rating:** 2
**Final Rating:** 3

**Summary:**

The paper applies flow matching with optimal transport to generate synthetic 3D craniofacial skeletal data from CT scans, aiming to address data scarcity and enable downstream tasks such as shape quality assessment and point–cloud–based shape completion. It voxelizes mesh data to train an OFTM‑based generator and evaluates realism using MSSIM, Improved/Probabilistic Precision and Recall, and performance on downstream tasks.

**Strengths:**

1. Tackles an important problem of realistic 3D craniofacial data generation, with clear relevance for reconstruction quality assessment and shape completion in clinical and surgical planning contexts.
2. Uses a modern flow‑matching with an optimal transport framework and reports multiple quantitative metrics (MSSIM, Improved Precision/Recall, Probabilistic Precision/Recall).
3. Includes downstream tasks (shape‑quality classification and point‑cloud–based shape completion) that move beyond pure sample visual inspection and start to probe the practical utility of the generated shapes.

**Weaknesses:**

1. The introduction is very high‑level and does not critically compare different generative families (autoregressive models, VAEs, GANs, diffusion/DDPM, flow/flow‑matching); there is no clear argument for why OFTM is particularly well-suited to this problem.
2. The motivation around “scarcity of data in medical image analysis” is underdeveloped and somewhat generic. It does not address concrete challenges, such as class imbalance from rare conditions, privacy/HIPAA constraints, heterogeneous image quality, or limited demographic coverage, nor why balanced or complete shapes are clinically necessary.
3. The related‑work section mostly enumerates methods without discussing their pros/cons, failure modes, or why alternatives (e.g., diffusion, VAEs, GANs) were not chosen, so the choice of OFTM is not convincingly justified.

**Detailed Comments:**

1. “different generative techniques such as Autoregressive models, Variational Autoencoders and Generative Adversarial Networks (GANs)”: add citations and run a spell/grammar check, especially in the introductory paragraph.
2. Strengthen the motivation section by explicitly discussing: why medical imaging datasets are often imbalanced (rare diseases, skewed severity distributions), privacy and regulatory constraints that limit data sharing, variable image quality and incomplete scans, restricted coverage across age, sex, and other demographics, and then explicitly linking these issues to the goals of balanced data generation and shape completion (e.g., handling corrupted scans or supporting pre‑operative planning).
3. In related work, move beyond listing prior techniques: summarize what each class of generative models does well and poorly for 3D medical shapes (e.g., mode collapse in GANs, blurry samples in VAEs, training cost of DDPMs, flexibility of flows) and explain the specific advantages of OFTM in this context (e.g., likelihood tractability, training stability, continuous‑time formulation).
4. Clarify why voxelization is chosen over direct mesh/point‑cloud processing. If there is a strong reason (e.g., compatibility with an off‑the‑shelf OFTM implementation), state it explicitly, and discuss trade‑offs in resolution, memory, and surface fidelity. Given that the downstream completion operates on point clouds, consider whether an entirely point‑cloud–based generative model would be more coherent.

**Justification Of Final Rating:**

Thank you to the authors for their detailed responses and clarifications. While the manuscript has been meaningfully revised in line with all the reviewer comments, I would like to comment specifically on the representation choice.

The statement that “voxelized representations are naturally produced by many medical imaging modalities such as CT, MRI, angiography, and PET” is correct at the raw‑image level, but it does not by itself justify voxelization for downstream shape‑based modeling. Voxel segmentations can be readily converted to surface meshes or point clouds (e.g., via marching cubes or sampling from the binary mask), and surface/point‑cloud representations are often more robust for geometric tasks because they (i) eliminate empty background volume, reducing memory and compute, (ii) better preserve sub‑voxel surface geometry once appropriately extracted, and (iii) avoid aliasing artefacts that arise from discretizing thin structures or oblique surfaces on a fixed grid. These advantages are one reason many recent works on 3D anatomical analysis, registration, and shape generation operate on meshes or point clouds rather than dense voxels.

That said, given the overall improvements to the manuscript and responses to other concerns, I am willing to update my recommendation from weak reject to borderline, while still encouraging the authors to reconsider or more thoroughly justify their reliance on voxelized representations for OFTM and provide a comparison with point cloud based shape generation and completion methods.

**Justification Of The Preliminary Rating:**

At this stage, the work appears technically solid and focused on a relevant application, but the motivation and methodological framework need substantial strengthening. The paper does not yet clearly argue why OFTM is the right choice compared with other established generative models, nor does it convincingly connect medical image analysis specific challenges to the proposed solution. The representational choice of voxelizing meshes is under‑justified given the later use of point clouds, and the benefits of synthetic data on downstream tasks are not fully quantified or dissected.

**Questions To Address In The Rebuttal:**

1. Conceptual choice: Why did you select OFTM instead of a DDPM, VAE, or GAN for 3D craniofacial shape generation? Are there specific properties (e.g., training stability, sample quality, likelihood estimation) that you empirically observed to be superior?
2. Data motivation: Beyond “data scarcity,” which specific challenges in your datasets (class imbalance, limited demographics, privacy constraints, corrupted scans) motivated the need for synthetic data and shape completion? Can you quantify any of these issues?
3. What is the rationale for voxelizing meshes rather than working directly with point clouds or meshes, especially since your downstream completion model uses point‑cloud representations? Did you compare voxel‑based versus point‑cloud–based generative models?
4. Are there qualitative clinical evaluations (e.g., expert rating of anatomical plausibility) or failure‑mode analyses for the generated shapes, particularly around regions critical for surgical planning?

---

> ### Author Response · Authors · 2026-01-25
>
> We thank the reviewer for their constructive feedback and for the insights on how to strengthen both the motivation and methodological framework. We have addressed the raised issues in the revised manuscript and provide detailed explanations below.
>
> ## Weaknesses
>
> 1. We revised paragraph 2 of the Introduction to better describe existing techniques and their limitations, clarifying why it is relevant to investigate OTFM in the medical domain. Since OTFM has demonstrated improved performance in generating natural images, we investigated the possibility of adapting this approach to 3D data and more importantly in the medical domain. In doing so, we find that OTFM does improve performance consistently across all metrics compared to DDPM, widely recognized SOTA paradigm in 3D medical image generation.
> 2. We thank the reviewer for the extensive suggestions on how to improve the motivation section. We revised paragraph 1 of the Introduction to motivate more clearly the reasons underlying the scarcity of medical data: we directly referenced the most prominent privacy regulation frameworks that limit data sharing; and introduced the issue related to the cost of annotating medical data. We also expanded the discussion on issues arising from dataset imbalance to include the factors highlighted by the reviewer and provided a more detailed explanation of how generative models can be used to mitigate these challenges.
> 3. We revised the Related work section to better articulate pros and cons of existing approaches. The section is now organized by first reviewing advances in 2D natural image generation and then discussing their application in 3D medical image generation. We also highlight the improvements brought by FM in 2D natural images, such as training stability, and conclude the section by clearly motivating our choice to investigate OTFM in 3D medical image generation and to compare it with diffusion-based approaches.
>
> ## Detailed Comments
>
> 1. We modified paragraph 2 of the Introduction and added the proper references.
> 2. The motivation section of the Introduction has been improved as described in the answer to Weakness 2.
> 3. We revised the Related work section as described in the answer to Weakness 3.
> 4. The rationale for using voxelized volumes is explained in the answer to Questions To Address In The Rebuttal 3.
>
> ## Questions To Address In The Rebuttal
>
> 1. We motivated our choice to investigate OTFM in the answer to Weakness 1. We empirically observed that OTFM is superior to DDPM in term of generation quality, as highlighted by both the quantitative results in Table 1 and the better performance in the downstream tasks. We also found that OTFM is more robust at inference time, as it results in less failed generations and in better conditioning with Quality Scores.
>
> 2. To broader the motivation section we revised paragraph 2 of the Introduction as described in the answer to Weakness 2. As regards our dataset, we added a sentence in the first paragraph of Section 3.3 to clarify that the main issue of our dataset is related to class imbalance. Figure 2 quantifies this issue by highlighting that QS 6 class contains far less samples (38) than other classes. We then leveraged synthetic data to augment and balance datasets, which we used to train models in two different clinical downstream tasks: Skull Alignment and Skull Completion.
>
> 3. The choice of voxelization over point cloud processing is motivated by the broader range of clinical tasks that can be addressed using voxelized volumes. Directly generating point clouds would limit the generality of our study to clinical tasks that specifically require point-cloud inputs. To preserve generality, we adopt a preprocessing pipeline based on voxelized representations, which are naturally produced by many medical imaging modalities such as CT, MRI, angiography, and PET.
>
>    Regarding the comparison between voxel-based and point-cloud–based generative models, we did not perform such an analysis. This is because the main focus of our work lies in the generative process, which is carried out in the latent space produced by the VQ-VAE. As the autoencoder is not the focus of our work, we opted to align our pipeline with the representations and practices most adopted in the relevant literature (Pinaya et al., ICCV 2022; Khader et al., Scientific reports 2023; Wang et al., IEEE Trans Med Imaging, July 2025).
>
> 4. Due to the limited time available during the rebuttal period, we were unable to involve physicians in a qualitative evaluation of the generated samples. Nevertheless, we conducted a detailed analysis of the synthetic datasets to more precisely assess the occurrence of failed generations and conditioning errors for FM and OTFM. The analyses confirms that OTFM yields more robust generation and better conditioning compared to DDPM. The results of this analysis have been added to the supplementary material, section D.

---

> ### Comment · Area_Chair_tr3k · 2026-02-01
>
> Please provide your final rating.
>
> AC

---

### Official Review · Reviewer_LaeL · 2026-01-09

**Confidence:** 4
**Preliminary Rating:** 4

**Summary:**

The sensitive nature of data and privacy limits access to large datasets in medical ML. This can be addressed through the use of generative synthetic data. This paper uses the Flow Matching with Optimal Transport to produce 3D craniofacial skeletal data, confirms it on skull alignment and shape reconstruction, and demonstrates that it is superior to DDPMs in quality and robustness.

**Strengths:**

This work is based upon Flow Matching with Optimal Transport (OTFM), a generative modelling technique introduced by Lipman et al. (2023).
This work uses 946 anonymized head CT scans, combining the public CQ500 dataset (355 scans) and a private Sant’Orsola Hospital dataset (591 scans).

**Weaknesses:**

A key weakness is the limited diversity and size of 3D medical data, with reliance on a single private clinical source, which may restrict generalization across populations, scanners, and anatomical variability.

**Detailed Comments:**

Nil

**Justification Of The Preliminary Rating:**

The preliminary rating is justified by the method’s clear novelty in extending OTFM to 3D medical data, strong experimental validation on clinical tasks, and consistent performance gains over established diffusion-based baselines.

**Questions To Address In The Rebuttal:**

Nil

---

> ### Author Response · Authors · 2026-01-25
>
> We appreciate the reviewer’s comments and their recognition of the strong experimental validation of the results on clinical downstream tasks. Regarding the limited size of the dataset used, as highlighted in the introduction, the creation of large and diverse medical datasets is challenging due to strict data sharing regulations and due to the high cost of annotating medical data. Still, our dataset comprises data from two different clinical sources, the public CQ500 dataset  and Sant’Orsola Hospital dataset, which results in an increased variety in scanners and anatomical variability.

---

> ### Comment · Area_Chair_tr3k · 2026-02-01
>
> Please provide your final rating
>
> AC

---

### Official Review · Reviewer_Ti1L · 2026-01-09

**Confidence:** 4
**Preliminary Rating:** 2
**Final Rating:** 3

**Summary:**

In this work, the authors trained a flow matching model (with optimal transport) to generate 3D craniofacial skeletal data. Then, they tested two downstream tasks, namely skull alignment and shape completion, to validate the generation ability. They showed that flow matching outperforms DDPM in both quality and robustness.

**Strengths:**

1. The work is solid; the details of the dataset and preprocessing are clearly described, the training procedure appears sound, and the evaluation goes beyond generated samples to include downstream tasks, which is particularly valuable for the medical domain.
2. Different experimental setups for data augmentation are included and compared.
3. Both quantitative results and visualizations of failure cases are presented and discussed, which is interesting to see.

**Weaknesses:**

MAJOR:
I feel that the main weakness of this submission lies in its presentation. Overall, the paper would benefit from more careful and polished writing before it can be considered ready for publication.
 In addition, some important information appears to be missing, which affects the clarity of the work.

1. Clarity: The paper lacks a clear and separate introduction to FM and OTFM. As these constitute the core methodology of the paper, I did not find sufficient background or explanation to fully understand them. In particular, FM seems to be skipped, with the paper directly introducing OTFM. It would be helpful to clarify the relationship and differences between these two methods.
2. Clarity: The description of conditioned FM based on the quality score is insufficient. In Section 3.2, the authors state that this will be described in Section 3.3, but I was unable to find such a description. Please correct me if this is due to my oversight. Additionally, including the conditioning mechanism in Figure 1 could improve the clarity of the presentation.
3. Presentation: In several places, the language lacks precision and rigor. As an example, I refer to paragraph 2 of the introduction.
* “among the most recent… LDMs, which aim at generating novel samples using DDPM.” The definition of LDM here seems inaccurate, as LDMs operate the diffusion process in latent space rather than directly on data space.
* “Still, the literature pertaining to image generation is rapidly evolving, and newer techniques have been proposed in the field of 2D natural image generation.” This sentence does not appear to add substantial value to the discussion.
* FM is not explicitly introduced, and the statement “provides better generation quality” is vague—better in what sense, and compared to which methods?
* The transition introduced by “however” feels abrupt, and the logical flow could be improved. The preceding sentence discusses the success of FM in 2D, followed by the statement that FM has not been applied to 3D generation. This transition would benefit from clearer motivation, for example by explaining why the lack of 3D FM models is unexpected or problematic. Moreover, it is unclear how the listed citations support the intended point.
----
MINOR (still weaknesses, but less critical):
1. The claim that “generative models could be used for unbalancing datasets” (last part of paragraph 1 in the introduction) would benefit from an appropriate reference.
2. Clarity: u_t is not explained in Equation 1, and epsilon is also not introduced.
3. It is unclear why the caption of Figure 1 refers to a “latent diffusion model”; it seems that FM would be more appropriate.
4. More details are needed on how features are extracted from Med3D.
5. Clarity: The “ref” setting does not appear to be described in the main text.
6. Regarding Table 1, the statement “Given the low cardinality of the test set, these values serve just as reference for analysing the other results” is unclear. Does the small test set affect the robustness of these results? If so, how are “the other results” related to or derived from them?
7. Reproducibility: The results appear to be based on a single run rather than multiple runs.
8. In Section 4.3, paragraph 2, it is unclear what the 50% and 25% refer to. Is the total number of samples the same across different settings?
9. The meaning of the “real” row in Table 2 is unclear. What does “real” represent in this context? isn't that real should be compared to the 3 scenarios, where you only use real images?

**Detailed Comments:**

The following points are intended as minor suggestions for improvement and should not be considered weaknesses. Addressing them could help improve the clarity and overall presentation of the final version.
1. When “ML” is first used in paragraph 1 of the introduction, the full term is not provided. While abbreviations may be introduced in the abstract, they should be defined again at their first occurrence in the main text. The same applies to the FID score and MSSSIM.
2. Regarding the quality score (QS), the definition of QS = 1 feels slightly unintuitive to me. Personally, defining QS = 1 as the reference case might be more natural, but this choice is ultimately up to the authors.
3. In Figure 3, it is unclear whether the two OTFM samples with QS4 also fail to depict the correct QS, as they are not highlighted with blue circles in the figure.
4. For Table 1 and Section 4.1, the presentation could potentially be clearer if the paper first states what aspect is being measured and then which metric is used, for example: realism – XX, diversity – YY, etc. Additionally, adding arrows to indicate the direction corresponding to better generation quality could further improve readability.

**Justification Of Final Rating:**

I appreciate the authors' effort in the rebuttal phase. I will raise my score from weak reject 2 to borderline accept 3, as my main concern -- the clarity of the paper --has been improved, though there is still room for further improvement.

**Justification Of The Preliminary Rating:**

As mentioned in the weaknesses above, I feel that the main issue of this submission lies in the lack of important information, such as the description of conditioned FM and the unexplained “ref” setting. In addition, the presentation, particularly in the introduction, could be improved. These issues could likely be addressed with more careful writing and clearer structuring.
At this stage, I am hesitant between a rating of 2 and 3. Based on the current version, it is difficult for me to recommend acceptance. However, I would be open to reconsidering my decision after the rebuttal if the authors are able to provide the missing information and clarify the points raised above.

**Questions To Address In The Rebuttal:**

1. How did you implement the conditioned FM?
2. Can you briefly explain the difference between FM and OTFM and why you choose the latter? Is there any literature on just FM?
3. What is the “ref” setting?
4. And what does the row of “real” represents in table 2?
5. And if you can, maybe consider a better introduction.

---

> ### Author Response · Authors · 2026-01-25
>
> We thank the reviewer for the thorough and detailed review and for the helpful suggestions on how to improve the structure and presentation of our study.
> # Weaknesses
> ## Major
> 1. We agree that the introduction to FM and OTFM was concise, mainly due to page limits. To improve clarity, we modified Sec. 3.1 to first introduce FM and to show how OTFM can be derived from it, highlighting the relationship between the two methods.
> 2. We understand that the previous description of the conditioning of the network through the Quality Score (QS) could create confusion. The conditioning mechanism was described in the supplementary, Sec. B.2, while in Sec. 3.3 we provided the definition of the QS. We felt that the reference to Sec. 3.3 was necessary, as we never mentioned the QS before, an important feature of our dataset. To avoid confusion, in the revised manuscript we added a description of the conditioning mechanism directly in Sec. 3.2 and reformulated the reference to Sec. 3.3 so that it is clearly linked to the presentation of the QS. In addition, we modified Fig. 1 to highlight the conditioning mechanism, as suggested.
> 3. We extensively revised the manuscript to improve clarity and presentation. The Introduction was rewritten to better motivate the need for 3D medical image generation and the relevance of investigating OTFM in this domain. The Related work section was reorganized to provide clearer context, and the Conclusion was revised to improve overall flow. In addition, we further improved the manuscript by addressing the issues highlighted by the reviewer as detailed below.
> ## Minor
> 1. We modified par. 1 of the Introduction to clarify why generative models can be used to balance datasets. Specifically, we emphasize that this is true only when the model is trained with conditioning.
> 2. We extended Sec. 3.1 to properly define epsilon and u_t.
> 3. The caption has been modified and extended to clarify the content of Fig. 1.
> 4. To explain how features are extracted from Med3D, we extended the relative description in par. 2 of Sec. 4.1.
> 5. To better understand the purpose of the “ref” setting, and to clarify how the metrics are computed, we modified Sec. 4.1 and 4.2. In Sec. 4.1, we specify that FID, precision, and recall for generative models are computed by comparing the real training set with the synthetic dataset. In Sec. 4.2, we explain that under the “ref” setting we report these metrics computed on real data. Specifically, FID, precision and recall are obtained by comparing the training and test sets of the real data, while MS-SSIM is computed on the real training set.
> 6. We agree that this sentence may cause confusion. It refers to the fact that when computing metrics based on features distributions (e.g., FID), real training data and synthetic data have the same cardinality, while in the “ref” results (e.g., train real data vs test real data) the cardinality differs. This setting reflects the one adopted by Pinaya et al. (ICCV 2022). To avoid confusion, we removed the sentence from the caption of Table 1.
> 7. We acknowledge the concern regarding reproducibility. Within the limited time of the rebuttal we managed to conduct an additional training run, which is consistent with our findings and indicates improved performance and robustness of OTFM compared to DDPM.
> ||FID|MS-SSIM|IP|IR|PR|
> |-|-|-|-|-|-|
> |DDPM|1.03|0.29|0.23|0.75|0.04|
> |OTFM|0.036|0.64|0.97|0.80|0.85|
> 8. We reformulated par. 2 of Sec. 4.3 to better define the different settings. Now each setting's name describes how the new dataset is made, and we specify its cardinality.
> The percentages refer to the portion of the real/synthetic set we are using to compose the new training set. The datasets across different settings contain different number of samples by design, as our goal is to measure the ability of synthetic data to augment existing datasets.
> 9. The tables showing the results on the downstream tasks have been refactored to reflect the updated formulation of the different settings and to clarify the meaning of the “real” row. In this context, “real” denotes that only real data have been used to train the downstream task model.
> # Detailed Comments
> 1  In the revised version abbreviations are correctly introduced.
>
> 3  In Fig. 3, QS 4 samples generated with OTFM are correctly conditioned as they contain a complete nasal area and do not include the mandible. Please note that we added a more precise description of the QSs in Sec. 3.3 to help better interpret their meaning.
>
> 4  We added arrows in Table 1, 5 and 6 to improve readability.
> # Questions To Address In The Rebuttal
> 1. We addressed the issue of model conditioning in the answer to Major Weakness 2.
> 2. Refer to the answer to Major Weakness 1.
> 3. The meaning of the “ref” setting has been explained in the answer to Minor Weakness 5.
> 4. Minor Weaknesses 8 and 9 answers clarify the meaning of the “real” row.
> 5. The introduction has been revised as highlighted in the answer to Major Weakness 3.

---

> > ### Comment · Reviewer_Ti1L · 2026-01-28
> >
> > Thanks for the reply and the revision. The clarity of the paper seems to have improved now.
> > I still have some follow-up questions about the new content.
> >
> > -----
> > Major weakness 1:
> >
> > * In the added content in sec 3.1, " An interesting case is the one that recovers DDPMs learning objective (Lipman et al., 2023; Esser et al., 2024), which means that FM subsumes DDPMs." Can you explain what does this mean?
> > * I also noticed the OTFM citation might be missing here
> >
> >
> > Major W 2:
> > * I'm trying to make sure I understand the setup correctly. From your previous manuscript, I had the impression that conditional FM was optional, but looking at the experiment section now, it seems like you're actually training and doing inference with the conditional version. Could you help clarify when you use the unconditional approach vs the conditional one?
> >
> > -----
> > Minor weakness 2: I'm still missing the reference for this claim.
> >
> > Minor weakness 5: Could you elaborate a bit about the ref? did you sample from the real data, if yes how? (No need to change the manuscript - just for my understanding)

---

> > > ### Author Response · Authors · 2026-01-28
> > >
> > > Thank you for the reply and for your positive feedback regarding the improved clarity. We provide answers to your follow-up questions below.
> > >
> > > ----
> > >
> > > Major weakness 1:
> > >
> > > - Flow Matching is a fully general method for data generation. Depending on how the forward process is defined (i.e. the choice of a_t and b_t in Eq. 2 of our manuscript) it can result in different generation paths.
> > >
> > >   OTFM is not a general method, but rather a specific instance of Flow Matching in which we define the path between samples of the two distribution as a straight line (Eq. 3).
> > >
> > >   Notably, a particular choice of a_t and b_t results in the DDPM path introduced by Ho et al. (2020). This is the reason why we say that FM subsumes DDPM.
> > >
> > >   In summary, FM is a general method that, depending on the definition of a_t and b_t, can recover OTFM, DDPM, and other generative processes, while OTFM and DDPM can both be interpreted as specific instances of FM.
> > >
> > >   Formal proofs of these relationships are provided in Section 4.1 of Lipman et al. (2023), as well as Sections 3 and 4 of Esser et al. (2024).
> > > - Thank you for the correction, we added the missing citation.
> > >
> > > Major weakness 2:
> > >
> > > - We always train and generate with conditioning. The reasons for doing so are threefold:
> > >
> > >   1. Training and generating with conditioning allow us to replicate the exact QS distribution of the training set in the generated data. Since the quantitative metrics used to evaluate the generated datasets (FID, precision, recall, MS-SSIM) are highly sensitive to the underlying distribution of the data, more similar distributions lead to more reliable metric computation and a more accurate assessment of model performance.
> > >   2. Being able to generate with conditioning allows to fully leverage the richness of the dataset. For instance, without conditioning it would be very difficult to generate QS 6 volumes, as their presence in the training set is very limited.
> > >   3. By generating with conditioning we are able to conduct the dataset balancing experiment, that otherwise would not be possible and has practical clinical relevance.
> > >
> > >   We decided to removed the sentence "The generation can be both conditioned on the Quality Score and unconditioned." in the last paragraph of section 3.2, as it may confuse readers about our setup, given that in our experiments we never generate without conditioning.
> > >
> > >   -----
> > >
> > > Minor weakness 1: in the newly revised manuscript we added an appropriate reference in the first paragraph of Section 1.
> > >
> > > Minor weakness 5: in the ref setting, we use all real training and test data to compute FID, precision and recall.
> > > In the case of FID, this is done as follows:
> > >
> > > 1. Extracting the features from all real data (train + test) using Med3D
> > > 2. Estimating training means and covariances using all training features
> > > 3. Estimating test means and covariances using all test features
> > > 4. Computing Fréchet distance between train and test distributions.
> > >
> > > In the case of MS-SSIM, we sample 1000 pairs from the training set and compute the metric for each pair. The result shown is the average of the 1000 measurements.

---

> > > > ### Comment · Reviewer_Ti1L · 2026-01-28
> > > >
> > > > Thanks for the quick reply. For major weakness 1, I got it now; thanks for the explanation. Somehow this sentence didn't really seem to be necessary for this paragraph (given that it's about the FM and OTFM, not really having that much to do with DDPM); that being said, I need to clarify that this is very minor. Honestly, there're quite a lot of other mentionings of DDPM that feel unnecessary (e.g. caption of Fig.1). For a clearer presentation, I would recommend the authors consider removing the unnecessary DDPM mentions.
> > > >
> > > > For major weakness 2, the clarification is clear now, and yes, if that is the case, the unconditional one should not be mentioned in the first place. I was wondering, does that also mean the FM and OTFM should be changed to conditional ones in Sec. 3?
> > > >
> > > > For minor W 5: I might not have expressed myself clearly enough before -- what I was wondering is that, it seems that in different settings you have different numbers of samples (sec 4.3). Did you sample the same number of samples for the FID score or?
> > > >
> > > >
> > > > -----
> > > >
> > > > Overall, I think my questions are addressed. My main concerns are all about clarity; and I can see that the clarity of the paper has been improved after the rebuttal phase.
> > > >
> > > > However, I'm concerned about the differences between the original manuscript and the rebuttal. For example, the first draft mentioned but not described the conditional FM (both in text and in figure 1), while the rebuttal claims that only conditional FM was used and the unconditional version was discarded. This inconsistency makes me feel the submission may not be ready yet.
> > > >
> > > > That being said, since my main questions have been resolved, I will raise my rating to borderline.

---

> > > > > ### Author Response · Authors · 2026-01-29
> > > > >
> > > > > Thank you for your comments. We try to address your remaining concerns below.
> > > > >
> > > > > ---
> > > > >
> > > > > Major Weakness 2: We believe that we should not change FM and OTFM to conditional ones in Section 3, as that section describes the theory behind FM and OTFM, which is the same independently of the presence of conditioning during training.
> > > > >
> > > > > ---
> > > > >
> > > > > Minor Weakness 5: When we compute the FID we use all the available samples. In all our experiments FID is computed following two possible configurations:
> > > > >
> > > > > 1. ref: the FID is computed between all the real data training samples (727 samples) and all the real data test samples (182 samples). In this setup we do not under-sample the real data training set to reach the same size as the real data test set. This is mainly because the FID computes means and covariances of the two distributions, therefore the higher the cardinality of the set, the more reliable the estimate is. Under-sampling a dataset would only reduce the reliability of the estimate.
> > > > >
> > > > > 2. All the FID metrics computed for DDPM and OTFM: since we can generate the same number of samples as the real data training set, we do so and compute the FID metrics between the real data training set (727 samples) and 727 generated samples.
> > > > >
> > > > > ---
> > > > >
> > > > > We would like to highlight that, in our opinion, there is no inconsistency between the original and revised manuscript. In particular, as for conditioned generation:
> > > > >
> > > > > TRAINING
> > > > >
> > > > > - In supplementary section B.2 of the original manuscript the conditioning mechanism used during training was already described in detail. The revised manuscript just includes the description from the supplementary material in the main text.
> > > > >
> > > > > - Neither in the first nor in the revised manuscript we mentioned unconditional training. On the other hand, we wrote in Section 3.2 "The model is conditioned on the Quality Score, a feature of our data described in Section 3.3.".
> > > > >
> > > > > INFERENCE
> > > > >
> > > > > - In Section 4.2 we wrote, "These models were used to synthesize datasets with the same cardinality and QS distribution as the original training set", implying that the generation is conditioned, as otherwise it would not have been possible to replicate the QS distribution of the real data training set.
> > > > >
> > > > > - In Section 3.2 we wrote that "The generation can be both conditioned on the Quality Score and unconditioned.", which may have caused confusion, as this possibility was not further explained nor used in the experiments. We removed the sentence to avoid confusion, but not to change the experimental setting. The possibility of generating without conditioning using our already trained model was true and remains true. This is a direct consequence of Classifier Free Guidance, as firstly explained in the paper by Ho and Salimans (2022).
> > > > >
> > > > > To summarize, the differences between the original and revised manuscript consist in having expanded the theoretical explanation on FM/OTFM and in having moved information from the supplementary material to provide more details on the training procedure for conditioned generation directly in the main paper.

---

> > > > > > ### Comment · Reviewer_Ti1L · 2026-01-31
> > > > > >
> > > > > > About the difference between the original manuscript and the rebuttal version on the topic of conditioning training: I did see the sentence in Section 3.2 from the first read; however, several places in the manuscript led me to think that both unconditional and conditional training were being used, as the conditional training was not well described at all. That's why I'd suggest considering explaining the conditional FM in the methodology part (I kinda of disagree that this is not needed in the main text; personal opinion, not necessarily to be weakness) and also adding the condition into Figure 1 (the previous Fig. 1 appears to show only unconditional).
> > > > > >
> > > > > > By the way, reviewers don't have to read the supplementary materials; on the top of that, I didn't see anywhere in the original manuscript pointing me to the supplementary. Appendix B.2 also has a very broad subtitle as "3d unet"—which doesn't directly relate to conditional FM training since UNet is just the backbone of the model; in principle it could be replaced with any backbone that would work.
> > > > > >
> > > > > > In short: it might be only me, but I found it quite difficult to understand from the original manuscript that only conditional training was used, and it gave me the impression that there was a difference with the rebuttal when you mentioned that unconditional training was never used.
> > > > > >
> > > > > > I will edit the part on justification of the final rating where I mentioned the difference between the original manuscript and the rebuttal, as it might not be totally fair.

---

### Author Rebuttal · Authors · 2026-01-25

**Rebuttal:**

We thank all the reviewers for their insightful comments. We appreciate the recognition of the soundness of the training procedure and of the experimental setup, as well as the positive feedback on the extensive quantitative and qualitative evaluation. We found the feedback on how to strengthen the presentation and improve the overall flow of the manuscript to be very helpful in improving our work. Detailed descriptions of our modifications and responses to the raised questions are provided in the official comments addressed to each reviewer.

We provide a revised version of the manuscript, with all modifications highlighted in red. Below, we summarize the changes:

- a substantial revision of the Introduction;
- reorganization and improvement of the Related work section;
- extension of Section 3.1 so as to introduce Flow Matching and Optimal Transport Flow Matching more formally and to clarify their relationship;
- update of Figure 1 and its caption to improve clarity in the description of the proposed architecture;
- addition of an explanation of the conditioning mechanism in the second paragraph of Section 3.2;
- extension of the description of the Quality Scores in Section 3.3;
- revision of Section 4.1 to improve overall clarity;
- revision of Section 4.2 to introduce the “*ref*” setting in the main text;
- in section 4.3, redefinition of the experimental settings for downstream task evaluation.
  Also, refactoring of the tables to improve readability;
- revision of the Conclusion to improve overall flow;
- addition of experiment to quantify failed generations and wrong conditioning in the supplementary, section D.

Update 28/01:

We provide a newly revised version of the manusript based on the [comment of Reviewer Ti1L](https://openreview.net/forum?id=O669OJ3fZf&noteId=oTdX0KrO6E). In addition to the above mentioned changes we modified the following:
- added relevant citations in paragraph 1 of the Introduction and paragraph 4 of Section 3.1;
- revised last paragraph of Section 3.2 to clarify the experimental setup.

**Supporting Material:**

/attachment/99394688145b39cff1b1aeb13f032f6ce90fcfb6.pdf

---

### Meta-Review · Area_Chair_tr3k · 2026-02-03

**Recommendation:** Accept (Poster)
**Confidence:** 3

**Metareview:**

All reviewers found that the presented results are convincing and properly evaluated. The main concern during the discussion phase, which was the clarity of the paper, was properly adressed. I recommend acceptance.

---

### Decision · Program_Chairs · 2026-02-13

Accept (Poster)